# Vitamin D Deficiency Prevalence in Hospitalized Patients with COVID-19 Significantly Decreased during the Pandemic in Slovakia from 2020 to 2022 Which Was Associated with Decreasing Mortality

**DOI:** 10.3390/nu15051132

**Published:** 2023-02-23

**Authors:** Juraj Smaha, Peter Jackuliak, Martin Kužma, Filip Max, Neil Binkley, Juraj Payer

**Affiliations:** 15th Department of Internal Medicine, Comenius University Faculty of Medicine, University Hospital, Ruzinovska 6, 826 06 Bratislava, Slovakia; 2Department of Pharmacology and Toxicology, Comenius University Faculty of Pharmacy, Odbojarov 10, 832 32 Bratislava, Slovakia; 3Department of Medicine, Geriatrics Faculty, Medical Sciences Center, University of Wisconsin, 1300 University Ave, Madison, WI 53706-1510, USA

**Keywords:** COVID-19, pneumonia, pandemic, vitamin D, 25-hydroxyvitamin D, SARS-CoV-2, vitamin D supplementation

## Abstract

The coronavirus disease 2019 (COVID-19) pandemic has led to changes in lifestyle, which could influence vitamin D status on a population level. The purpose of our study was to compare 25-hydroxyvitamin D (25[OH]D) levels in patients hospitalized because of severe COVID-19 during two waves of the pandemic (2020/21 vs. 2021/22). A total of 101 patients from the 2021/22 wave were compared with 101 sex- and age-matched subjects from the 2020/21 wave. Patients from both groups were hospitalized during the winter season from 1 December to 28 February. Men and women were analyzed together and separately. The mean 25(OH)D concentration increased from 17.8 ± 9.7 ng/mL to 25.2 ± 12.6 ng/mL between waves. The prevalence of vitamin D deficiency (<20 ng/mL) decreased from 82% to 54%. The prevalence of adequate serum 25(OH)D concentration (>30 ng/mL) increased from 10% to 34% (*p* < 0.0001). The proportion of patients with a history of vitamin D supplementation increased from 18% to 44% (*p* < 0.0001). Low serum 25(OH)D concentration was independently associated with mortality after adjusting for age and sex for the whole cohort of patients (*p* < 0.0001). The prevalence of inadequate vitamin D status in hospitalized patients with COVID-19 in Slovakia decreased significantly, probably due to a higher rate of vitamin D supplementation during the COVID-19 pandemic.

## 1. Introduction

In December 2019, several cases of viral pneumonia of unknown etiology emerged in the city of Wuhan, Hubei province, China [1]. In the same month, a new viral pathogen—severe acute respiratory syndrome coronavirus 2 (SARS-CoV-2)—was discovered, and a new disease, coronavirus disease 2019 (COVID-19), was identified. The virus spread quickly worldwide, and on 11 March 2020, the World Health Organization declared COVID-19 a pandemic [2].

During the first half of 2020, almost all countries started implementing safety measures, such as social distancing, stay-at-home orders, the closing of non-essential facilities, and a ban on traveling to halt spreading of the virus. While necessary for constraining the virus, possible adverse effects of these measures on lifestyle, eating habits, physical activity, mood, and social life have been discussed. For example, Ammar et al. showed that home confinement during the COVID-19 pandemic negatively affected all levels of physical activity and led to more time spent sitting. Additionally, an unhealthy pattern of food consumption was exhibited [3]. A small study showed that serum markers such as glucose, total cholesterol, and LDL increased post-lockdown even in previously healthy young adults [4].

Vitamin D was widely discussed during the pandemic by medical professionals, as well as by the lay public. Several studies pointed toward the possible adverse effects of vitamin D deficiency on mortality and disease severity in COVID-19 [5]. Sources of vitamin D include cutaneous synthesis from cholecalciferol upon ultraviolet B radiation exposure, diet (e.g., cod, liver, salmon, egg yolk, or beef liver), and supplementation. However, sunlight exposure is the predominant source of vitamin D [6]. Many speculated that changes in eating habits and, more importantly, less time spent in sunlight during lockdown could negatively influence vitamin D status on a population level [7].

Several studies, predominantly in younger patients and children, compared serum levels of 25(OH)D in the first months of the pandemic with pre-pandemic levels, with inconclusive results. Yu et al. And, similarly, Rustecka et al. showed that the COVID-19 pandemic restrictions led to decreased serum 25(OH)D levels among pediatric populations [8,9]. On the other hand, Meoli et al. did not show a higher prevalence of vitamin D insufficiency among late adolescents during the pandemic [10].

Contrary to the negative results of lifestyle changes and lower sunlight exposure to vitamin D status, the rising awareness of the potential harmful effects of vitamin D deficiency could lead to higher use of vitamin D supplements among the general population [11].

In our previous work dealing with changes of 25(OH)D in hospitalized patients with COVID-19 between the end of December 2021 and the beginning of January 2022 [12], we noticed that the average 25(OH)D concentration in patients admitted to the hospital was significantly higher than we observed in the previous period. We speculated that during the pandemic, after the end of stay-at-home orders, the lack of sunlight did not have such a profound effect on the concentration of vitamin D status. Alternatively, increasing awareness of the potential beneficial effects of vitamin D on immune functions during the pandemic could have led to higher use of supplementation in the general public, with a positive net effect on vitamin D status.

The aim of the present study was to compare the serum concentrations of 25(OH)D between the second (2020/2021) and the third wave (2021/2022) of the pandemic in hospitalized patients with COVID-19 in Slovakia.

## 2. Materials and Methods

We analyzed patients hospitalized in the internal medicine department of University Hospital Bratislava, Ruzinov, during the second (Group 1) and the third (Group 2) wave of the COVID-19 pandemic. Patients from both waves were hospitalized during the winter season: the second wave was considered from 1 December 2020 to 28 February 2021, and the third wave was considered from 1 December 2021 to 28 February 2022. During the second wave, a total of 2696 COVID-19 patients were hospitalized at the University Hospital Bratislava, Ruzinov. During the third wave, a total of 860 COVID-19 patients were hospitalized at our facility. A total of 101 (61 males/40 females; 12% from all hospitalized COVID-19 patients at our facility) patients from the third wave of the pandemic fulfilled our inclusion criteria and were compared to 101 (61 males/40 females; 4% from all hospitalized COVID-19 patients at our facility) sex- and age-matched subjects from the second wave of the COVID-19 pandemic. Patients were first matched for sex, then for age ±1 year. If several options were available for the match, patients with the closest value of BMI were chosen.

A total of 202 patients (102 males/100 females) fulfilling inclusion criteria were included in this study. The inclusion criteria were as follows:COVID-19 pneumonia was the primary diagnosis upon admission;A severe COVID-19 infection was present;The presence of SARS-CoV-2 was detected by a reverse transcriptase–polymerase chain reaction (RT–PCR) using a nasopharyngeal swab;Serum 25(OH)D levels were obtained precisely at admission.

Severe COVID-19 was defined as clinical signs of pneumonia and one of the following: respiratory rate > 30 breaths per minute, severe respiratory distress; or oxygen saturation < 90% on room air [13].

Our facility’s laboratory did not routinely perform epidemiological surveillance using whole-genome sequencing (WGS). The Public Health Authority of the Slovak Republic launched a systematic national epidemiological surveillance using WGS in selected laboratories from 1 March 2021. From March 2021 until the end of June 2021, the most prevalent variant of concern detected was Alpha (B.1.1.7). Delta variants (B.1.617.2) were present in the Slovak population until the end of 2021. The Omicron variant appeared rapidly at the beginning of 2022 and continued to be prevalent until March 2022, with dominant lineages BA.2 and BA.2.9 [14].

Demographic characteristics, comorbidities, hematological and biochemical laboratory results on admission, and information regarding the intensity of oxygen therapy were collected from electronic medical records and discharge summaries by two physicians using a standardized approach.

All patients included in the study received six milligrams of intravenous dexamethasone daily, according to the standard of care. All patients with oxygen therapy via a high-flow nasal cannula or invasive mechanical ventilation received six milligrams of dexamethasone plus one of the following: anakinra subcutaneously (100 mg twice daily for three days, followed by 100 mg daily for seven days), tocilizumab intravenously (8 mg/kg actual body weight administered as a single i.v. dose), or baricitinib orally (4 mg once daily up to 14 days, dose adjusted according to the actual eGFR) in the second wave, and baricitinib (4 mg once daily up to 14 days, dose adjusted according to the actual eGFR) orally during the third wave. All patients admitted up to January 2021 were supplemented with vitamin D during hospitalization according to the following scheme per local protocol in University Hospital Bratislava: loading dose 30,000 IU of cholecalciferol daily for the first three days, followed by 7500 IU cholecalciferol per day. After January 2021, the local treatment protocol was updated, and vitamin D supplementation during hospitalization was no more part of the standard of care in our institution. 

Serum 25(OH)D concentrations (in ng/mL) were obtained on admission using an automated electrochemiluminescence system (Eclesys Vitamin D Total II, 2019, Roche Diagnostics GmBH, Mannheim, Germany) with repeatability < 20 ng/mL SD ≤ 1.1 ng/mL; >20 ng/mL coefficient of variation ≤ 5.5% and intermediate precision < 20 ng/mL SD ≤ 1.4 ng/mL; and >20 ng/mL coefficient of variation ≤ 7.0%. The serum 25(OH)D detection limit was 3 ng/mL [15]. All patients in the study were assessed with the same vitamin D detection method, and there was no change in the 25(OH)D measurement method between the waves. Analyses were performed in a laboratory, which was part of an external quality assessment system from accredited groups SEKK Czech Republic; NEQAS, GenQA Great Britain; and INSTAND, RfB Germany [16].

A serum 25(OH)D concentration > 30 ng/mL was considered vitamin D sufficiency; a concentration between 20 and 30 ng/mL was considered vitamin D insufficiency, and vitamin D deficiency was defined as a serum 25(OH)D concentration < 20 ng/mL in accordance with existing guidelines [17].

Vitamin D status seems to be sex-related [18]. Therefore, both sexes were analyzed together, as well as separately. Analyses were also performed according to the age of the participants. Patients were divided into two groups according to age. Younger age was defined as a chronological age < 65 years, and older age was defined as a chronological age of 65 and more [19]. Both sexes were analyzed in both age groups, together and separately.

For statistical analysis of continuous variables, an unpaired *t*-test of mean values was used, and for analysis of categorical variables, a chi-square test of independence was used. For analysis of vitamin D serum level categories (sufficiency, insufficiency, deficiency) in the whole cohort and age and sex groups, a chi-square test with contingency tables was used. The mean serum 25(OH)D levels were compared in the whole cohort and in sex and age categories using an unpaired *t*-test of mean values. The relationship between serum 25(OH)D concentration and mortality adjusted for sex and age was assessed in the whole cohort. Logistic binary regression analysis with death as a dependent variable was used. Statistical analyses were performed using the SPSS program (ver. 21.0; IBM Corp., Armonk, NY, USA). The *p*-value < 0.05 was considered to be statistically significant.

## 3. Results

A total of 101 sex- and age-matched patients in each wave (61 males, 40 females, mean age 69 years) were analyzed. Baseline clinical and laboratory characteristics between the waves (second wave—Group 1; third wave—Group 2) are displayed in Table 1.

The mean concentration of 25(OH)D on admission during the second wave of the pandemic (Group 1) was 17.8 ng/mL, which increased to 25.2 ng/mL during the third wave (Group 2) (*p* < 0.0001). On admission, 82% of patients from Group 1 were 25(OH)D deficient, and 10% were 25(OH)D sufficient. In Group 2, 54% of patients were 25(OH)D deficient, and 34% of patients were 25(OH)D sufficient (*p* < 0.0001). The proportions of patients regarding vitamin D cutoff values in both groups are displayed in Figure 1A. There was no difference in the prevalence of major comorbidities except for chronic kidney disease, which was frequently observed in Group 1 (*p* < 0.001), and for dementia, which was more frequent in Group 2 (*p* = 0.02). The major comorbidities associated with a cardiopulmonary reserve—chronic heart failure, chronic pulmonary disease, anemia and concomitant pulmonary embolism—did not differ significantly between groups. The proportion of patients with a history of vitamin D supplementation increased from 18% to 44% (*p* < 0.0001).

Changes in vitamin D concentrations in both sexes were also analyzed separately (Table 2).

In the population of males, the proportion of vitamin-D-deficient patients decreased from 84% to 48%, and the proportion of vitamin-D-sufficient patients increased from 8% to 37% (*p* < 0.0001) (Figure 1C). The mean 25(OH)D concentration in males increased by 9.1 ng/mL, from 17.2 ng/mL to 26.3 ng/mL (*p* < 0.0001). In females, the proportion of vitamin-D-deficient patients decreased from 80% to 62%, and the proportion of vitamin-D-sufficient patients increased from 13% to 28% (*p* = 0.29) (Figure 1B). The mean 25(OH)D concentration in females increased by 4.9 ng/mL, from 18.7 ng/mL to 23.6 ng/mL (*p* = 0.07).

The prevalence of vitamin D deficiency decreased in younger as well as in older patients (Table 3). In patients < 65 years old, the proportion of vitamin-D-deficient patients decreased from 81% to 44%, and the proportion of vitamin-D-sufficient patients increased from 3% to 34%. In older patients (>65 years), the prevalence of vitamin D deficiency decreased by 24%, and the prevalence of vitamin D sufficiency increased by 20% (Figure 2).

In men, a statistically significant decrease of vitamin-D-deficient patients was observed in both age groups; in younger (< 65 years) males by 44%, and in older males (> 65 years) by 29% (*p* < 0.001 and *p* < 0.002, respectively). In women, the prevalence of vitamin-D-deficient patients decreased in both age groups, although the difference was not statistically significant. In older females (> 65 years,), the prevalence of vitamin D deficiency decreased from 79% to 60%, and vitamin D sufficiency increased from 15% to 31%, which was borderline statistically significant (*p* = 0.056) (Table 3).

The most significant absolute change of 25(OH)D concentration between waves was observed in younger males (10.7 ng/mL, *p* < 0.002), and the smallest absolute change of 25(OH)D concentration was observed in younger females (2.5 ng/mL, *p* = 0.68) (Figure 3). Except for younger females, in all other groups, a statistically significant increase of mean 25(OH)D concentration was observed between waves (see Table 3 and Figure 3).

Regarding markers of inflammation, the highest numbers of monocytes and lowest numbers of lymphocytes were observed in Group 2. There was no difference between the numbers of neutrophils and C-reactive protein between both groups (Table 1).

In Group 2, a slight reduction of mortality of 6% was observed, which was not statistically significant (*p* = 0.58). Binary logistic regression analysis performed on the whole cohort (all patients admitted during two COVID-19 waves) showed that there was a significant association of 25(OH)D concentration with mortality, even after adjusting for age and sex (Table 4).

An increase in serum 25(OH)D concentration of one ng/mL leads to approximately a 7% increase in the chance of survival (Figure 4).

## 4. Discussion

In this study, we found a significant reduction in the prevalence of vitamin D deficiency among hospitalized patients with COVID-19 between the second and third waves of the pandemic. The prevalence of vitamin-D-deficient patients decreased by 28%, and the prevalence of vitamin D sufficiency increased by 24%. This change in vitamin D status coincided with a more than doubling of the proportion of patients taking vitamin D supplementation. Moreover, an increase in mean 25(OH)D concentration was observed for both men and women regardless of age group, except for females younger than 65. These findings are surprising given the high prevalence of vitamin D deficiency in Europe, and likely even higher in Eastern Europe [20]. The increase of serum 25(OH)D levels by one ng/mL was associated with a ~7% reduction in mortality for the whole cohort of patients (both groups combined).

To date, several studies have investigated the changes in vitamin D status during the COVID-19 pandemic; the majority focused on people under the age of 18 years [21,22]. A meta-analysis of five studies comprising 4141 people under the age of 18 showed significantly lower serum 25(OH)D levels during the COVID-19 pandemic compared with pre-pandemic years [23]. These lower serum 25(OH)D levels were not observed among infants (under one year), where either no change [9] or even an increase [8] of 25(OH)D concentration was observed. This could have been the result of regular vitamin D supplementation in this age group and more attentive caretaking of the youngest children during the COVID-19 pandemic [9].

Concerns about a decrease in the serum concentration of vitamin D due to pandemic measures have not yet been confirmed in an adult population. Two studies from Northern Italy did not find a clinically relevant impact on vitamin D status from confinement during the first year of the COVID-19 pandemic (January–December 2020) compared with pre-pandemic years [24,25]. Lippi et al. observed an increase in serum 25(OH)D, accompanied by a reduced prevalence of 25(OH)D deficiency during the COVID-19 lockdown (March to May 2020), followed by a slight reduction of median serum 25(OH)D levels in the post-lockdown period (May to December 2020) [24]. In the study from South Korea on adults aged 19 years and older, which also included measurements of vitamin D from the second year of the pandemic (measurements up to November 2021), a significant increase in serum 25(OH)D concentration was observed in females, as well as in males. Contrary to our results, this increase in 25(OH)D serum levels was more significant in females, especially in elderly females [26].

Articles emphasizing the potential beneficial role of vitamin D supplementation in preventing and ameliorating COVID-19 infection were widely discussed and cited during the pandemic [27]. Indeed, several meta-analyses showed that vitamin D deficiency was associated with a worse prognosis and mortality of COVID-19 pneumonia, although with a high risk of bias and heterogeneity in multiple observational studies [28,29]. Some authors advocated supplementation with higher than commonly recommended doses of vitamin D during the pandemic to achieve and sustain serum 25(OH)D concentrations above 50 ng/mL [30]. Even some governmental agencies endorsed vitamin D supplementation, addressing the concerns about the potential worsening of musculoskeletal health on a populational level during the pandemic [31]. All this could have led to a higher awareness of the possible extraskeletal effects of vitamin D in the field of research and the general public.

Somagutta et al. analyzed people’s micronutrient searches using an online platform. Vitamin D searches rose eight-fold in 2020–2021 from 2004, while nearly doubling throughout 2019–2021. This was probably due to curiosity about the effectiveness of vitamin D during the COVID-19 pandemic and could be translated into vitamin D supplement usage [11].

In a trend analysis of laboratory-based 25(OH)D samples comparing the yearly average of 25(OH)D in the 12 months before the onset of the COVID-19 pandemic with the first 12 months of the pandemic in Ireland, a yearly mean 25(OH)D concentration increase of 1.1 ng/mL/year was observed. If the 25(OH)D duplicate was selected as the last in sequence for the trend analysis in that study, then the average 25(OH)D increase was even higher at 2 ng/mL/year [32]. At the same time, the dose of new-to-market vitamin D supplements increased significantly during the pandemic, with an increase in the frequency of supplements exceeding the upper intake level and the maximum safe level. The prevalence of patients with serum 25(OH)D levels above 50 ng/mL increased substantially, which concerned the study’s authors [32]. No case of vitamin D hypervitaminosis was seen in our cohort of COVID-19 patients, and only three patients exceeded the 50 ng/mL cutoff level.

Vitamin D supplementation for the prevention of acute respiratory tract infections repeatedly demonstrates a significant overall protective effect of this intervention compared with placebo control. Martineau et al. showed that patients who were very vitamin D deficient and those not receiving bolus doses experienced the most benefit [33]. Jolliffe et al. showed that patients receiving regular doses of vitamin D (400–1000 IU) for up to 12 months and those with younger ages benefited most [34]. Regarding COVID-19, a recent systematic review and meta-analysis showed that dietary supplementation with vitamin D was associated with a significantly lower risk of COVID-19 severity and mortality [35]. Several meta-analyses of randomized controlled trials indicated a beneficial role of vitamin D supplementation on ICU admission [36] and mortality [37]. At the same time, some did not prove any effect of supplementation in COVID-19 patients [38].

In the present study, the main aim was to compare changes in vitamin D status between COVID-19 waves in Slovakia. However, the outcome regarding vitamin D status on admission was also evaluated. A slight insignificant decrease in mortality in Group 2 was observed. At the same time, for the whole cohort of patients (both Group 1 and 2 combined), an independent inverse relationship between serum 25(OH)D levels at the time of admission and mortality was detected. The major limitation regarding mortality was that the treatment protocol changed considerably during the COVID-19 pandemic at our institution, and patients from different COVID waves were treated differently. For example, only the patients from Group 1 were supplemented with vitamin D during hospitalization. Vitamin D is a threshold nutrient, so patients with a severe deficiency will most likely benefit from supplementation [39]. Specifically, regarding COVID-19 disease, Gibbons et al. showed that patients with the lowest levels of 25(OH)D (0–19 ng/mL) exhibited the most significant decrease in COVID-19 infection following supplementation [40]. We could speculate that some severely 25(OH)D deficient patients in Group 1 could have improved their nutritional status upon supplementation and thus could have exhibited a milder course of the disease and lower mortality. Similarly, there was a difference in the treatment strategy with anti-inflammatory agents between waves. While in Group 1, patients on HFNV received predominantly anakinra and tocilizumab, these treatments were largely unavailable during the third wave of the pandemic in Slovakia. Patients with HFNV in Group 2 were treated predominantly with baricitinib instead. Thus, changes of the treatment strategy could have had a major impact on disease outcomes.

Some authors found a significant inverse relationship between low 25(OH)D and inflammatory markers in COVID-19 [41], while others did not [42]. In the present study, values of CRP and neutrophils did not differ significantly between groups despite significantly increased mean 25(OH)D levels in Group 2. It can be argued that the effect of vitamin D on the course of COVID-19 is mediated by antimicrobial peptides like cathelicidin, which cannot be assessed by standard serum inflammatory biomarkers like CRP or IL-6. Interestingly, in Group 2, a significantly higher number of monocytes in peripheral blood was present. An active form of vitamin D can induce the proliferation of monocytes. It can improve macrophage function, such as phagocytosis, chemotaxis, and production of cathelicidins, thus ultimately modulating the innate and adaptive immune response [43,44]. The bioavailability of 25(OH)D to macrophages is a crucial determinant of the physiological control of its immune response [44]. Monocytes and macrophages are linked to the heterogeneity of the SARS-CoV-2 infection course and, depending on the signals from the microenvironment (e.g., vitamin-D-receptor-related signaling), could be either friends or foes in COVID-19 [45,46].

Serum 25(OH)D levels could drop rapidly with the onset of acute inflammatory illness, suggesting that inflammation can affect 25(OH)D metabolism in various ways [47,48,49]. Our previous work from a real clinical practice showed that serum 25(OH)D levels decreased significantly in patients with COVID-19 pneumonia during the first 48 h after hospital admission. The absolute 25(OH)D change between hospital admission and day 4 was 4.8 ng/mL [12]. Smolders et al. showed experimentally that serum 25(OH)D levels decreased within hours of initiating a systemic inflammatory response. Thus, patients who were ill for a longer period before hospitalization could have had lower 25(OH)D levels upon admission [50]. Whether low 25(OH)D in COVID-19 reflects functional vitamin D deficiency linked to the worse prognosis or represents only a laboratory phenomenon remains to be found in adequately designed randomized trials of vitamin D supplementation.

High levels of misinformation exposure were observed during the pandemic, with 73% of people reporting some exposure to misinformation about COVID-19 vaccination. Exposure to misinformation was directly correlated with vaccine hesitancy [51]. Similarly, it must be noted that the potential immunomodulatory effects of vitamin D have often been overestimated, and the results of studies were misinterpreted during the COVID-19 pandemic. Of great concern is that misleading sources also suggested or directly stated there was no evidence to support COVID-19 public health prevention measures and, at the same time, stated that vitamin D had preventative or curative abilities against COVID-19 [52].

During the second wave of the pandemic, vaccination was not widely available in Slovakia. The vaccination program started on 26 December, and at the end of February 2021 only 6.46% of the population had received the first dose of the vaccine, most of whom were healthcare workers and other first responders. This changed considerably throughout 2022, and by the end of February 2022 approximately 46% of people had received at least one dose [53]. However, more than 70% of hospitalized patients during the third wave of the COVID-19 pandemic were unvaccinated despite Slovakia’s widely available COVID-19 vaccination at that time. We can hypothesize that the observed significantly higher vitamin D levels resulted partly from alternative “immune boosting” strategies not endorsed by major medical entities [54,55].

Our study has several limitations. We evaluated a relatively small group of patients. It is a single-center study within a specific geographic area; thus, results cannot be widely generalizable to the populations of other geographical regions. The exact dose and duration of vitamin D supplementation before hospitalization were unknown. Serum 25(OH)D concentrations in winter are generally about 50–70% of summertime values, and there is evidence that the 25(OH)D accumulates in skeletal muscle cells, which provide a functional store during the winter months [56]. However, we only knew patients’ values of serum 25(OH)D at the beginning of the hospitalization, i.e., during winter months.

Interestingly, a significantly higher prevalence of patients with chronic kidney disease was observed in Group 1. This was probably caused by our institution’s triage policy during the second wave when COVID-19 patients were admitted to the hospital according to the major comorbidities they had at the time of infection (e.g., a patient with COVID-19 with severe kidney disease was sent to the internal medicine department). This practice changed during the third wave when the general COVID-19 ward was established. Nevertheless, the kidneys play an important role in vitamin D metabolism and regulation of its circulating levels. The progression of chronic kidney disease is associated with lowering 25(OH)D serum levels [57]. Serum 25(OH)D is bound to vitamin D binding globulin, which is filtered in the glomerulus and then reabsorbed in the proximal tubules by binding to megalin and cubilin receptors [58]. With CKD progression, associated proteinuria and decreased megalin activity could lead to renal wasting of a considerable amount of vitamin D and VDBP, resulting in more profound vitamin D deficiency [59]. This renal wasting of vitamin D could be exacerbated during acute inflammation [49] and is of particular interest in the population of patients with COVID-19.

Our study also has several strengths. To the best of our knowledge, this is the first study comparing changes in serum 25(OH)D concentration in hospitalized patients with COVID-19 between selected waves during the pandemic. Potential confounders of vitamin D deficiency did not significantly affect our results because patients were sex- and age-matched. There was also no significant difference in BMI between groups, and venous samples were taken during the same season of the year.

## 5. Conclusions

In conclusion, our study showed that the prevalence of vitamin D deficiency in patients hospitalized because of COVID-19 decreased significantly during the 12 months between the second and the third wave of the pandemic in Slovakia. The prevalence of vitamin D sufficiency increased both in males and females, although only in males was this change statistically significant. The mean 25(OH)D concentration increased by 7.45 ng/mL/year. The most significant absolute change was observed in younger males and the smallest in the cohort of young females. The inverse relationship between vitamin D serum levels and mortality from COVID-19 was detected. Further research in trend analysis of yearly changes of 25(OH)D serum concentration before, during, and after the COVID-19 pandemic is indicated on a broader population level.

## Figures and Tables

**Figure 1 nutrients-15-01132-f001:**
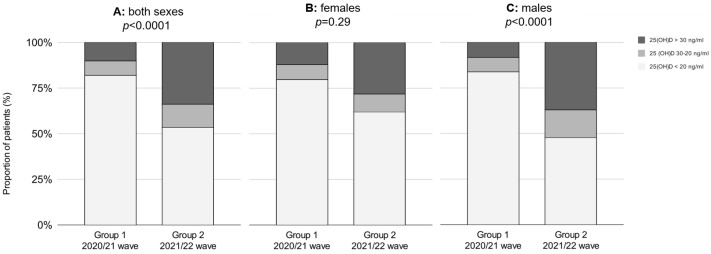
The proportion of patients regarding 25(OH)D cutoff values between the waves in both sexes (**A**), females (**B**) and males (**C**), respectively. Group 1—second wave (2020/21); Group 2—third wave (2021/22).

**Figure 2 nutrients-15-01132-f002:**
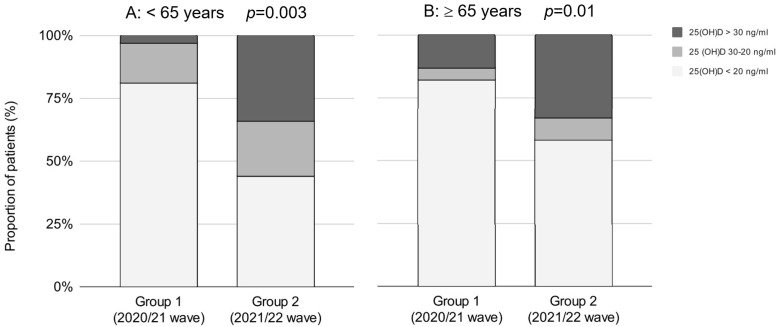
The proportion of patients regarding 25(OH)D cutoff values between the waves in patients <65 years old (**A**) and patients ≥65 years old (**B**), respectively. Group 1—second wave (2020/21); Group 2—third wave (2021/22).

**Figure 3 nutrients-15-01132-f003:**
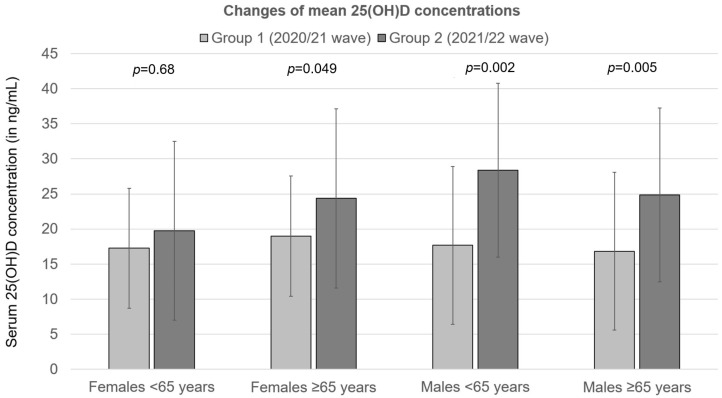
Changes of mean 25(OH)D concentrations in patients hospitalized for severe COVID-19 between the second (2020/21) and the third (2021/22) wave of the pandemic. Group 1—second wave (2020/21); Group 2—third wave (2021/22).

**Figure 4 nutrients-15-01132-f004:**
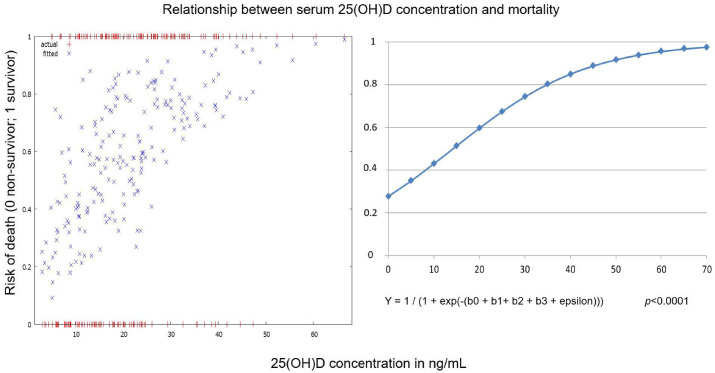
Relationship between serum 25(OH)D concentration upon hospital admission and mortality for the whole cohort of hospitalized patients with COVID-19 pneumonia (patients from both waves analyzed together). Y = dependent variable (risk of death), b1 = independent variable (age), b2 = independent variable (sex), b3 = independent variable (serum25(OH)D concentration).

**Table 1 nutrients-15-01132-t001:** Baseline clinical and laboratory characteristics on admission between patients admitted to the hospital in the second wave (2020/21) (Group 1), and patients admitted to the hospital in the third wave (2021/22) (Group 2) of the pandemic. Data are expressed as numbers and percentages or means ± standard deviations.

Variable	Group 1 (*n* = 101)	Group 2 (*n* = 101)	*p* Value
Age (years)	70 ± 14	70 ± 14	N/A
Males/females, *n* (%)	61 (60%)/40 (40%)	61 (60%)/40 (40%)	N/A
Survivors/Non-survivors, *n* (%)	57 (56%)/44 (44%)	63 (62%)/38 (38%)	0.58
BMI (kg/m^2^)	29 ± 5.8	29 ± 6.7	0.96
Vitamin D supplementation before hospitalization, *n* (%) Duration of dyspnea before hospitalization (days)	18 (18%)	44 (44%)	<0.0001
25(OH)D (ng/mL)	17.8 ± 9.7	25.2 ± 12.6	<0.0001
Vitamin D sufficiency, *n* (%) *	10 (10%)	34 (34%)	<0.0001
Vitamin D insufficiency, *n* (%)	8 (85%)	13 (12%)	<0.0001
Vitamin D deficiency, *n* (%)	83 (82%)	54 (54%)	<0.0001
Arterial hypertension, *n* (%)	73 (72%)	75 (74%)	0.87
Diabetes mellitus, *n* (%)	42 (42%)	41 (41%)	0.91
Chronic heart failure, *n* (%)	15 (15%)	25 (25%)	0.11
Chronic pulmonary disease, *n* (%)	14 (14%)	18 (18%)	0.37
Concomitant pulmonary embolism, *n* (%)	10 (10%)	13 (13%)	0.43
Anemia, *n* (%)	23 (23%)	25 (25%)	0.62
Dementia, *n* (%)	13 (13%)	28 (28%)	0.01
Cirrhosis, *n* (%)	1 (1%)	3 (3%)	0.32
Chronic kidney disease, *n* (%)	35 (35%)	13 (13%)	0.001
High flow oxygen, *n* (%)	45 (45%)	63 (63%)	0.07
Invasive mechanical ventilation, *n* (%)	5 (5%)	8 (8%)	0.41
Leukocytes (10 × 9/L)	9.1 ± 4.7	11 ± 6.5	0.04
Neutrophils (10 × 9/L)	8.4 ± 8.7	10 ± 12	0.24
Lymphocytes (10 × 9/L)	1.1 ± 1.3	0.6 ± 0.5	<0.0001
Monocytes (10 × 9/L)	0.6 ± 1.1	1.1 ± 0.6	<0.0001
C-reactive protein (mg/L)	124 ± 84	139 ± 90	0.22

* Vitamin D sufficiency is defined as 25(OH)D concentration > 30 ng/mL; vitamin D insufficiency is defined as 25 (OH)D concentration between 20 and 30 ng/mL; vitamin D insufficiency is defined as 25(OH)D concentration < 20 ng/mL.

**Table 2 nutrients-15-01132-t002:** Comparison of vitamin D status upon admission between men and women admitted to the hospital in the second wave (Group 1), and in the third wave (Group 2) of the COVID-19 pandemic. Data are expressed as numbers and percentages or means ± standard deviations.

Variable	Group 1	Group 2	*p* Value
Men	N = 61	N = 61	
25(OH)D (ng/mL)	17.2 ± 8.6	26.3 ± 12.8	<0.0001
Vitamin D sufficiency, *n* (%) *	5 (8%)	23 (37%)	
Vitamin D insufficiency, *n* (%)	5 (8%)	9 (15%)	<0.0001
Vitamin D deficiency, *n* (%)	51 (84%)	30 (48%)	
Women	N = 40	N = 40	
25(OH)D (ng/mL)	18.7 ± 11.3	23.5 ± 12.4	0.07
Vitamin D sufficiency, *n* (%) *	5 (13%)	11 (28%)	0.29
Vitamin D insufficiency, *n* (%)	3 (7%)	4 (10%)	
Vitamin D deficiency, *n* (%)	32 (80%)	24 (62%)	

* Vitamin D sufficiency is defined as 25(OH)D concentration > 30 ng/mL; vitamin D insufficiency is defined as 25 (OH)D concentration between 20 and 30 ng/mL; vitamin D insufficiency is defined as 25(OH)D concentration < 20 ng/mL.

**Table 3 nutrients-15-01132-t003:** Comparison of vitamin D status upon admission according to age group in patients admitted to the hospital in the second wave (Group 1), and in the third wave (Group 2) of the COVID-19 pandemic. Data are expressed as numbers and percentages or means ± standard deviations.

Sex	Age Group	Vitamin D Status	Group 1	Group 2	*p* Value
Females	<65 years	25(OH)D (ng/mL)	17.3 ± 8.6	19.8 ± 12.8	0.68
		Vitamin D sufficiency, *n* (%) *	0 (0%)	1 (14%)	0.68
		Vitamin D insufficiency, *n* (%)	1 (14%)	1 (14%)	
		Vitamin D deficiency, *n* (%)	6 (86%)	5 (72%)	
	≥65 years	25(OH)D (ng/mL)	19 ± 8.6	24.4 ± 12.8	0.049
		Vitamin D sufficiency, *n* (%)	5 (15%)	10 (31%)	0.056
		Vitamin D insufficiency, *n* (%)	2 (6%)	3 (9%)	
		Vitamin D deficiency, *n* (%)	26 (79%)	19 (60%)	
Males	<65 years	25(OH)D (ng/mL)	17.7 ± 11.3	28.4 ± 12.4	0.002
		Vitamin D sufficiency, *n* (%)	1 (4%)	10 (40%)	0.001
		Vitamin D insufficiency, *n* (%)	4 (16%)	6 (24%)	
		Vitamin D deficiency, *n* (%)	20 (80%)	9 (36%)	
	≥65 years	25(OH)D (ng/mL)	16.8 ± 11.3	24.8 ± 12.4	0.005
		Vitamin D sufficiency, *n* (%)	4 (11%)	13 (35%)	0.002
		Vitamin D insufficiency, *n* (%)	1 (3%)	3 (8%)	
		Vitamin D deficiency, *n* (%)	31 (86%)	21 (57%)	

* Vitamin D sufficiency is defined as 25(OH)D concentration > 30 ng/mL; vitamin D insufficiency is defined as 25 (OH)D concentration between 20 and 30 ng/mL; vitamin D insufficiency is defined as 25(OH)D concentration < 20 ng/mL.

**Table 4 nutrients-15-01132-t004:** Results of logistic binary regression analysis with death as a dependent variable.

Variable	Coefficient	Standard Error	t	*p* Value
Constant	−0.457	0.437	−1.045	0.29
Age	−0.052	0.013	−4.050	<0.0001
Sex	−0.414	0.336	−1.230	0.22
25(OH)D (ng/mL)	0.067	0.017	4.023	<0.0001

## Data Availability

The data presented in this study are available on request from the corresponding author.

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
