# Peer review of "Vitamin D Deficiency Prevalence in Hospitalized Patients with COVID-19 Significantly Decreased during the Pandemic in Slovakia from 2020 to 2022 Which Was Associated with Decreasing Mortality"

_nutrients, 2023, doi:10.3390/nu15051132_

Round 1
Reviewer 1 Report
The manuscript shows a decrease in vitamin D deficiency and an improvement in outcome for patients admitted to hospital in consecutive winters and waves of covid-19. It is a relatively straightforward study with carefully age and sex matched groups of patients, strengthening the results. A few additional confounders are not really discussed, or not mentioned at all. Their impact is likely minor, but they should still be mentioned.
Materials and Methods
Why 101 patients, and what fraction of the total admissions does this represent in each year? Is it the maximum number that could be age and sex matched or is there another reason for the choice of 101 (e.g. considered representative of the population)?
Lines 95-108: these differences in treatment will not affect the main aim of the paper, to explore vitamin D status on admission, but may impact on the analysis of disease outcome. Presumably baricitinib was shown to be the most effective treatment and so was used exclusively in Group 2, possibly giving them an outcome advantage. The vitamin D supplementation of Group 1 may have improved their outcome beyond what it might have been without the supplementation. These points should be mentioned somewhere – either at the appropriate point in the Results/Discussion or as a limitation of the work.
Lines 109-113: give the uncertainty in the vitamin D system results, and also details of any QA for the lab where the results were obtained (e.g. DEQAS) – or lack thereof.
Results
Lines 150-152: Could chronic kidney disease impact 25(OH)D levels? Comment on this.
Figures 1 and 2: while the figures provide easy visual interpretation, the data is already provided in tables so they are somewhat repetitive. Consider combining Figures 1 and 2 (i.e. add the columns for the entire groups to Figure 2, which then becomes Figure 1).
Lines 217-219: provide some further explanation of the significance (or not) of these observations for the severity of disease. Or put another way, why did you include this data – what does it indicate?
Line 220-230 and Line 295-297: these are associated with the comment on lines 95-108. These aspects should be linked and discussed.
Line 226, and 241: Figure 5 and the ~7% change for every 1 ng/ml 25 (OH)D comes from analysis of both groups together (all ages and both sexes), so you cannot say the outcome is for a given age and sex category as this was not analysed.
Lines 307-310 What was the matching % of patients who had been vaccinated in Group 1? Probably zero as vaccination was in its infancy at this time, but clarify the position for Slovakia.
Also some comment should be made on the variant of covid-19 most prevalent in the two winters, and the more general expectations of outcome for the variants e.g. was omicron most prevalent in Group 2?
Minor points
Please check all non-integer numbers and ensure a full stop/period not a comma is used e.g. 2.5 not 2,5. Start with the abstract.
Line 50: ultraviolet B radiation (it is not light, which refers to the visible part of the spectrum).
Line 222: a not s
Changes in line spacing e.g. line 200, and the Conclusions
Minor language editing needed, though this does not impact on understanding.
Author Response
Dear Reviewer
We thank you for the thoughtful reviews, suggestions, and comments to improve our manuscript “Vitamin D deficiency prevalence in hospitalized patients with COVID-19 significantly decreased during the pandemic in Slovakia from 2020 to 2022”. We have reviewed whole manuscript and rewritten according to comments.
All changes in the manuscript are highlighted with yellow color.
Our point-by-point responses follow:
The manuscript shows a decrease in vitamin D deficiency and an improvement in outcome for patients admitted to hospital in consecutive winters and waves of covid-19. It is a relatively straightforward study with carefully age and sex matched groups of patients, strengthening the results. A few additional confounders are not really discussed, or not mentioned at all. Their impact is likely minor, but they should still be mentioned.
Point 1.: Why 101 patients, and what fraction of the total admissions does this represent in each year? Is it the maximum number that could be age and sex matched or is there another reason for the choice of 101 (e.g. considered representative of the population)?
Response: 101 patients was the maximum number of patients fulfilling our inclusion criteria in the third wave. We also added fractions of the total admissions at our facility during respective COVID waves. See lines 79-89.
Point 2.: Lines 95-108: these differences in treatment will not affect the main aim of the paper, to explore vitamin D status on admission, but may impact on the analysis of disease outcome. Presumably baricitinib was shown to be the most effective treatment and so was used exclusively in Group 2, possibly giving them an outcome advantage. The vitamin D supplementation of Group 1 may have improved their outcome beyond what it might have been without the supplementation. These points should be mentioned somewhere – either at the appropriate point in the Results/Discussion or as a limitation of the work.
Response: These treatment differences are now discussed in length in the Discussion section, see lines 313-333.
Point 3: Lines 109-113: give the uncertainty in the vitamin D system results, and also details of any QA for the lab where the results were obtained (e.g. DEQAS) – or lack thereof.
Response: This was updated accordingly; see lines 127-130 and 132-134.
Point 4: Lines 150-152: Could chronic kidney disease impact 25(OH)D levels? Comment on this.
Response: A paragraph about the possible impact of CKD is now part of the Discussion; see lines 390-398.
Point 5: Figures 1 and 2: while the figures provide easy visual interpretation, the data is already provided in tables so they are somewhat repetitive. Consider combining Figures 1 and 2 (i.e. add the columns for the entire groups to Figure 2, which then becomes Figure 1).
Response: Figures were updated accordingly.
Point 6: Lines 217-219: provide some further explanation of the significance (or not) of these observations for the severity of disease. Or put another way, why did you include this data – what does it indicate?
Response: A paragraph discussing a relationship between COVID-19 severity, biomarkers of inflammation, and vitamin D is now part of the "Discussion"; see lines 333-347.
Point 7: Line 220-230 and Line 295-297: these are associated with the comment on lines 95-108. These aspects should be linked and discussed.
Response: According to your suggestion, this is now mentioned in the Discussion section, see lines 313-333, mainly from line 318.
Point 8: Line 226, and 241: Figure 5 and the ~7% change for every 1 ng/ml 25 (OH)D comes from analysis of both groups together (all ages and both sexes), so you cannot say the outcome is for a given age and sex category as this was not analysed.
Response: These lines were rewritten for clarity.
Point 9: Lines 307-310 What was the matching % of patients who had been vaccinated in Group 1? Probably zero as vaccination was in its infancy at this time, but clarify the position for Slovakia.
Response: As you rightly pointed out, the % of fully vaccinated adults was zero at that time; the Slovakian position was clarified and is now part of the Discussion, see lines 367-374.
Point 10: Also some comment should be made on the variant of covid-19 most prevalent in the two winters, and the more general expectations of outcome for the variants e.g. was omicron most prevalent in Group 2?
Response: A paragraph about COVID variant prevalence among the Slovakian population was added; see lines 102-108
Minor points
All formal, minor points were updated accordingly.
Reviewer 2 Report
In this work it is described for the first time the vitamin D scenario during Covid-19 two waves between 2020 and 2022.
Analysing vitamin D levels in Covid-19 positive hospitalized patients, authors have understood that vitamin D levels are higher during the second wave of pandemia both in males and females and that there is an inverse correlation between vitamin D levels and mortality for Covid-19.
I think that this work is well written and data well described, in particular figures are well created.
Author Response
Dear Reviewer
We thank you for the thoughtful and encouraging review to our manuscript “Vitamin D deficiency prevalence in hospitalized patients with COVID-19 significantly decreased during the pandemic in Slovakia from 2020 to 2022”.
All changes in the manuscript are highlighted with yellow color.
Reviewer 3 Report
Dear Authors,
It is a nice well-presented manuscript. But I would like to suggest you few changes. I have summarized them below;
1. The severity of COVID-19 you have mentioned based on respiratory rate and oxygen saturation. Was CT scan of lungs, not performed? The best indicator for severity assessment would be the percentage of the pulmonary area affected.
2. Secondly you have not mentioned if the selected patients had any cardio-pulmonary co-morbidities further leading to a rapid deterioration or the need of oxygen. You have mentioned chronic heart failure, but what about a lung disease or anemia etc.
3. How was their diet, were they having a vit D issues even before the virus hit them?
4. Were the patients asked to fill in a questionnaire stating there food habits, life conditions referring to sunlight exposure etc.?
5. You have not mentioned if the testing the levels of vit D was a routine hospital protocol or it was amended due to covid or it was the author’s choice. If it was your choice how were the funds arranged?
6. The lot of patients selected by you for the study is very vague. I would suggest you to reconsider and state the inclusion and exclusion criteria more specific
7. The patient outcomes and prognosis are not clear, just focusing on the levels of vit D is a very subjective thing.
8. Nor any details of their endocrinological state is stated
I would like you to revise the article on the above specified points.
All the best!
Author Response
Dear Reviewer
We thank you for the thoughtful reviews, suggestions, and comments to improve our manuscript “Vitamin D deficiency prevalence in hospitalized patients with COVID-19 significantly decreased during the pandemic in Slovakia from 2020 to 2022”. We have reviewed whole manuscript and rewritten according to the comments.
All changes in the manuscript are highlighted with yellow color.
Our point-by-point responses follow:
Reviewer 3
Dear Authors,
It is a nice well-presented manuscript. But I would like to suggest you few changes. I have summarized them below;
Point 1: The severity of COVID-19 you have mentioned based on respiratory rate and oxygen saturation. Was CT scan of lungs, not performed? The best indicator for severity assessment would be the percentage of the pulmonary area affected.
Response: The severity criteria were based on previously published guidelines established by The Surviving Sepsis Campaign (SSC) and the National Institutes of Health. Since neither of those guidelines requires a CT scan for severity assessment, we did not include a CT scan in severity assessment in our study. We agree that adding these data would improve the manuscript. However, given the lack of resources in our facility at that time, the CT scan was performed only for some patients upon admission.
Point 2: Secondly, you have not mentioned if the selected patients had any cardio-pulmonary co-morbidities further leading to a rapid deterioration or the need of oxygen. You have mentioned chronic heart failure, but what about a lung disease or anemia etc.
Response: We reviewed patients' health records for any other cardiopulmonary conditions, which are now part of Table 1 and briefly discussed in the results section see lines 174-176.
Point 3: How was their diet, were they having a vit D issues even before the virus hit them?
Response: Unfortunately, we do not have data about the dietary style of patients in the present study, nor do we know the 25(OH)D serum levels before hospitalization.
Point 4: Were the patients asked to fill in a questionnaire stating there food habits, life conditions referring to sunlight exposure etc.?
Response: We did not give patients a questionnaire stating food habits and sunlight exposure upon admission. Patients were asked about regular medications and supplements (including those prescribed by physicians and others obtained over the counter) as a part of their standard medical history.
Point 5: You have not mentioned if the testing the levels of vit D was a routine hospital protocol or it was amended due to covid or it was the author's choice. If it was your choice how were the funds arranged?
Response: It is not a standard practice in our facility (University Hospital Bratislava) to obtain serum 25(OH)D levels upon admission. It was our choice at the internal medicine department to evaluate serum 25(OH)D levels in hospitalized COVID-19 patients. We are allowed to order such a study from our laboratory in a hospitalized patient once a month (the insurance company covers it).
Point 6: The lot of patients selected by you for the study is very vague. I would suggest you to reconsider and state the inclusion and exclusion criteria more specific
Response: The paragraph about inclusion criteria was rewritten for clarity; see lines 92-98. Regarding the inclusion criteria, we consider our criteria valid for discussing vitamin D and COVID-19. The emphasis is on (1) COVID-19 as a primary diagnosis upon admission, (2) the severity and specificity of the disease (hypoxemia and pulmonary form), and (3) assessment of 25(OH)D strictly upon admission in all patients (instead of previous 25OHD values months or even years before the onset of COVID-19, which may not reflect actual 25OHD levels upon hospitalization)
Point 7: The patient outcomes and prognosis are not clear, just focusing on the levels of vit D is a very subjective thing.
Response: The present study was not explicitly designed to address mortality; however, we found it important to point out that vitamin D is inversely associated with mortality. The major limitation regarding the present study's outcome is that treatment protocol has changed considerably during the COVID-19 pandemic in our country, and patients from different COVID waves were treated differently. However, the primary aim of our study was to assess and discuss changing trends of serum 25(OH)D levels of hospitalized patients with COVID-19, not outcome and prognosis.
Point 8: Nor any details of their endocrinological state is stated
Response: Unfortunately, we did not have data about the hormonal profile of patients included in the study. During clinical nad basic laboratory evaluation, there were no signs of major endocrinologic issues at that time in our patients.
Reviewer 4 Report
The fact that serum 25(OH)D concentrations drops rapidly with the onset of acute inflammatory illness should be discussed. As a result, serum 25(OH)D concentration at time of hospital admission with COVID-19 is more indicative of risk of adverse effects than of pre-disease concentration.
Letter to the Editor: Vitamin D deficiency in COVID-19: Mixing up cause and consequence.
Smolders J, van den Ouweland J, Geven C, Pickkers P, Kox M.Metabolism. 2021 Feb;115:154434. doi: 10.1016/j.metabol.2020.154434.
Serum 25-hydroxyvitamin D Concentration Significantly Decreases in Patients with COVID-19 Pneumonia during the First 48 Hours after Hospital Admission
J Smaha, M Kužma, P Jackuliak, S Nachtmann, F Max… - Nutrients, 2022 - mdpi.com
… Disease 2019 (COVID-19) affects 25-hydroxyvitamin D (25[OH]D) concentration. The
objective of our study was to examine serum 25(OH)D levels during COVID-19 pneumonia.
Conclusions: Serum 25-(OH)D is a negative acute phase reactant, which has implications for acute and chronic inflammatory diseases. Serum 25-(OH)D is an unreliable biomarker of vitamin D status after acute inflammatory insult. Hypovitaminosis D may be the consequence rather than cause of chronic inflammatory diseases.
Waldron J.L., Ashby H.L., Cornes M.P., Bechervaise J., Razavi C., Thomas O.L., Chugh S., Deshpande S., Ford C., Gama R. Vitamin D: A negative acute phase reactant. J. Clin. Pathol. 2013;66:620–622. doi: 10.1136/jclinpath-2012-201301. - DOI - PubMed
Bang U.C., Novovic S., Andersen A.M., Fenger M., Hansen M.B., Jensen J.-E.B. Variations in Serum 25-Hydroxyvitamin D during Acute Pancreatitis: An Exploratory Longitudinal Study. Endocr. Res. 2011;36:135–141. doi: 10.3109/07435800.2011.554937. - DOI - PubMed
Bertoldo F., Pancheri S., Zenari S., Boldini S., Giovanazzi B., Zanatta M., Valenti M.T., Carbonare L.D., Cascio V.L. Serum 25-hydroxyvitamin D levels modulate the acute-phase response associated with the first nitrogen-containing bisphosphonate infusion. J. Bone Miner. Res. 2010;25:447–454. doi: 10.1359/jbmr.090819. - DOI - PubMed
Krishnan A., Ochola J., Mundy J., Jones M., Kruger P., Duncan E., Venkatesh B. Acute fluid shifts influence the assessment of serum vitamin D status in critically ill patients. Crit. Care. 2010;14:R216. doi: 10.1186/cc9341.
COVID-19 cases and deaths in Slovakia for the second and third waves can be found st
https://www.worldometers.info/coronavirus/country/slovakia/
A quick estimate is that there were 2000 cases/day and 50 deaths/day during the first wave
And 5000 cases/day and 25 deaths/day during the second wave.
It is it assumed that the virulence of the SARS-CoV-2 virus can be estimated from the ratio of deaths/cases, the ratios are 0.025 in the second wave and 0.005 in the third wave.
Going back to the effect of COVID-19 reducing serum 25(OH)D concentration, COVID-19 might have only 20% of the power to reduce 25(OH)D in the third wave compared to the second wave.
On the other hand, the high CRP values for both waves as shown in Table 1 suggest that the virulence of the SARS-CoV-2 virus was little changed between the two waves.
Another thing to consider is the mean serum 25(OH)D concentrations in summer and winter for inhabitants of Slovakia of the age of those studied in this work.
Serum 25(OH)D concentrations in winter are generally about 50-70% of summertime values
25-hydroxyvitamin D, IGF-1, and metabolic syndrome at 45 years of age: a cross-sectional study in the 1958 British Birth Cohort.
Diabetes. 2008 Feb;57(2):298-305. doi: 10.2337/db07-1122.
Temporal relationship between vitamin D status and parathyroid hormone in the United States.
PLoS One. 2015 Mar 4;10(3):e0118108. doi: 10.1371/journal.pone.0118108.
due to recycling of 25(OH)D stored in muscles.
The Role of Skeletal Muscle in Maintaining Vitamin D Status in Winter. Curr Dev Nutr. 2019 Jul 25;3(10):nzz087. doi: 10.1093/cdn/nzz087.
Skeletal Muscle and the Maintenance of Vitamin D Status.Nutrients. 2020 Oct 26;12(11):3270. doi: 10.3390/nu12113270.
Thus, there are a number of issues that need to be addressed before the claim can be made that
The prevalence of inadequate vitamin D status in hospitalized patients with COVID-19 in Slovakia decreased 25 significantly due to a higher rate of vitamin D supplementation during the COVID-19 pandemic.
Another thing to investigate is the vitamin D doses taken by the patients in the second wave. Such data do not seem to be reported in the manuscript. I suggest that the patients be contacted and asked how much vitamin D they were taking.
Another possibility is to review records of serum 25(OH)D concentrations for non-COVID-19 patients for both waves.
Suggest citing:
Association between vitamin D supplementation and COVID-19 infection and mortality.
Sci Rep. 2022 Nov 12;12(1):19397. doi: 10.1038/s41598-022-24053-4.
For Figure 5, please show the regression fit to the data and provide the equation and p value.
Please use this form throughout: ng/mL
Significant digits. The general rule is that no more non-zero digits should be given than are justified by the uncertainty of the value.
See "Too many digits: the presentation of numerical data"
https://www.ncbi.nlm.nih.gov/pmc/articles/PMC4483789/
If the uncertainty is greater than about 7%, only two non-zero digits are justified.
P values should be given to two decimal places unless the first two are 00 or the number lies between 0.045 and 0.054. If the first two are 00, then only one non-zero digit can be given.
Thus,
Please adjust P values as stated above.
|
Age (years) |
69.67±14.48 |
Should be
|
Age (years) |
70±14 |
|
Leukocytes (10x9/L) |
9.10±4.73 |
Should be
|
Leukocytes (10x9/L) |
9.1±4.7 |
Please review all numbers in abstract, text, tables, and figures and adjust accordingly.

Author Response
Dear Reviewer,
We thank you for the thoughtful reviews, suggestions, and comments to improve our manuscript “Vitamin D deficiency prevalence in hospitalized patients with COVID-19 significantly decreased during the pandemic in Slovakia from 2020 to 2022”.
All changes in the manuscript are highlighted with yellow color.
Our responses follow:
Thank you very much for your valuable points. The relationship between acute inflammation and lower 25(OH)D levels is now discussed in the Discussion section; see lines 345-358. As highlighted by the reviewer, serum 25(OH)D concentration at the time of hospital admission with COVID-19 is more indicative of the risk of adverse effects than pre-disease concentration, which is why we included only patients with serum 25(OH)D levels taken at the time of hospital admission. One additional paragraph discussing inflammatory biomarkers and COVID-19 was added; see lines 334-347.
Although we very much appreciate your suggestions, we are, at this time, unfortunately, unable to provide a control group of non-COVID-19 patients matched both for age and sex in the respected period of the year. We understand that serum 25(OH)D concentrations are generally about 50-70% of summertime value. Thank you for stressing out that 25(OH)D concentration is maintained during winter months, probably by reuptake by skeletal muscles. This is now included in our Limitations section; see lines 380-384. However, we only knew patients' values of serum 25(OH)D at the beginning of the hospitalization. We could contact only a fraction of patients, understanding that many of them died because of COVID-19, and for many of them, we do not simply have a phone contact.
We provided a control group of non-COVID outpatients evaluated for routine internal control at our facility during fall and winter in respective COVID waves. We were able to identify 126 outpatients with serum 25(OH)D levels evaluated during September 2020 to March 2021 and 292 outpatients with serum 25(OH)D levels evaluated during the same period one year later (2021/2022 wave). Quick results are summarized in the following table:

Given the fact that these are collected for a longer span of time, data are not sex and age-matched, the sample for the control group is considerably small, and the mean age of outpatients is approx 13 years lower than in our study group; we are unsure, whether this data would provide more insight into the present study.
All forms were corrected to ng/mL. All p values and all numbers in the abstract, tables, text, and figures were updated according to your suggestions.